# The Chemical Composition of Oils and Cakes of *Ochna serrulata* (Ochnaceae) and Other Underutilized Traditional Oil Trees from Western Zambia

**DOI:** 10.3390/molecules26175210

**Published:** 2021-08-27

**Authors:** Adela Frankova, Anna Manourova, Zora Kotikova, Katerina Vejvodova, Ondrej Drabek, Bozena Riljakova, Oldrich Famera, Mbao Ngula, Mukelabai Ndiyoi, Zbynek Polesny, Vladimir Verner, Jan Tauchen

**Affiliations:** 1Department of Food Science, Faculty of Agrobiology, Food and Natural Resources, Czech University of Life Sciences Prague, Kamycká 129, 16500 Praha-Suchdol, Czech Republic; frankovaa@af.czu.cz (A.F.); riljakova@af.czu.cz (B.R.); famera@af.czu.cz (O.F.); 2Department of Crop Sciences and Agroforestry, Faculty of Tropical AgriSciences, Czech University of Life Sciences Prague, Kamýcká 129, 16500 Praha-Suchdol, Czech Republic; manourova@ftz.czu.cz (A.M.); polesny@ftz.czu.cz (Z.P.); 3Department of Chemistry, Faculty of Agrobiology, Food and Natural Resources, Czech University of Life Sciences Prague, Kamýcká 129, 16500 Praha-Suchdol, Czech Republic; kotikova@af.czu.cz; 4Department of Soil Science and Soil Protection, Faculty of Agrobiology, Food and Natural Resources, Czech University of Life Sciences Prague, Kamýcká 129, 16500 Praha-Suchdol, Czech Republic; vejvodova@af.czu.cz (K.V.); drabek@af.czu.cz (O.D.); 5ProjectEDUCATE, P.O. Box 910316, Mongu, Zambia; nmn4869@gmail.com; 6School of Natural Resources, University of Barotseland, P.O. Box 910316, Mongu, Zambia; mukelabai.ndiyoi@gmail.com; 7Department of Economics and Development, Faculty of Tropical AgriSciences, Czech University of Life Sciences Prague, Kamýcká 129, 16500 Praha-Suchdol, Czech Republic; vernerv@ftz.czu.cz

**Keywords:** oil-bearing plants, underused crops, *Ochna serrulata*, *Schinziophyton rautanenii*, *Parinari curatellifolia*, sustainability

## Abstract

Currently, the negative effects of unified and intensive agriculture are of growing concern. To mitigate them, the possibilities of using local but nowadays underused crop for food production should be more thoroughly investigated and promoted. The soybean is the major crop cultivated for vegetable oil production in Zambia, while the oil production from local oil-bearing plants is neglected. The chemical composition of oils and cakes of a three traditional oil plant used by descendants of the Lozi people for cooking were investigated. *Parinari curatellifolia* and *Schinziophyton rautanenii* oils were chiefly composed of α-eleostearic (28.58–55.96%), linoleic (9.78–40.18%), and oleic acid (15.26–24.07%), whereas *Ochna serrulata* contained mainly palmitic (35.62–37.31%), oleic (37.31–46.80%), and linoleic acid (10.61–18.66%); the oil yield was high (39–71%). *S. rautanenii* and *O. serrulata* oils were rich in γ-tocopherol (3236.18 μg/g, 361.11 μg/g, respectively). The *O. serrulata* oil also had a very distinctive aroma predominantly composed of *p*-cymene (52.26%), *m*-xylene (9.63%), γ-terpinene (9.07%), *o*-xylene (7.97), and limonene (7.23%). The cakes remaining after oil extraction are a good source of essential minerals, being rich in N, P, S, K, Ca, and Mg. These plants have the potential to be introduced for use in the food, technical, or pharmaceutical industries.

## 1. Introduction

The botanical diversity of vascular plants is estimated to be as high as 250,000–300,000 species [1]; however, only about 200 species make a major contribution to food production [2]. Currently, some 75% of the world’s food is generated from only twelve plants, including rice, maize, and wheat [3]. Locally varied indigenous food production systems are thus at significant risk of being lost due to the expansion of intensive agriculture and concentration on a single crop. A similar trend can be seen in the loss of traditional knowledge, culture, and skills of indigenous peoples. With this decline, agrobiodiversity is also disappearing as well as many traditionally consumed edible species (including varieties), whose existence is considered to be threatened or that are already extinct [4]. Particularly in developing countries, these consequences can ultimately lead to food insecurity and the use of unsafe food sources [5].

Zambia is one of the third-world countries where food security and safety are significantly worsened by agrobiodiversity loss. Parts of Zambia, especially the western province (formerly Barotseland), now suffer from hidden hunger and limited accessibility to good-quality food [6,7]. High-yielding crops such as maize, rice, cassava, mangos, tomatoes, cashew nuts, and groundnuts that require a lot of water and fertilizer are often substituted for traditional food plants and wild foods. Because of the semiarid climate in the western province, conditions for cultivation of high-value, resource-demanding species are challenging to achieve (personal communication, 28 November 2018). Given the proximity to the botanically rich South African floristic regions, Zambia has a diverse range of flowering plants covering more than 3500 species [8]. Despite this fact, the local people seem to overlook the nutritional and commercial potential in these traditionally used species, and the knowledge about these local fruits, vegetables, and oil crops is rapidly disappearing (Polesny, Z., unpublished data). The vast majority of these species have not been described by means of modern scientific methods [9]. Hence, the identification and documentation of traditional foods from developing countries is currently of the utmost interest. Undoubtedly it can lead to advantages to both developed and developing countries in terms of economic potential from their commercialization as food or food supplements. This can be demonstrated by açaí fruit (*Euterpe oleracea*) and camu camu (*Myrciaria dubia*). Those plants, traditionally used in South America, remained unknown to the western market until the 1990s, but nowadays are sold worldwide as rich sources of vitamin C and other antioxidants [10].

Nowadays the market with edible oils in Zambia is increasing; however, the local production covers only one-third of the market demand. The major cultivated oil-bearing plants are soybean, cotton seed and sunflower, accounting for 60, 19 and 15% of local production, respectively. Interestingly, due to the lack of knowledge about the cultivation of soybean, the plant is not very popular among farmers, and its potential remains underdeveloped [11]. Statistics about the oils from local plants are not available, but their use in rural areas is still current. Information gathered during a visit to the western Zambian province revealed that for food preparation people prefer to use oils obtained from the seeds of three local plants, namely *Schinziophyton rautanenii*, *Parinari curatellifolia* and *Ochna serrulata*, and thus, species usually used for other purposes. For example, the fruit of *S. rautanenii*, also known as manketti or mongongo, is reported to be an important food source for many communities in Namibia [12]. The most valuable is the kernel, from which oil and flour are obtained. The oil is rich in conjugated fatty acids, while the kernel flour is rich in essential amino acids [13]. The oil is not prone to oxidation; thus, the kernel can be stored for long time [14] and plays an important role in times of food shortage, when the nuts are ground and made into porridge (personal communication, 5 June 2019). Despite the oil being used locally for food preparation, commercially it is employed only in cosmetic products, especially in South Africa. *P. curatellifolia*, called mubula in the Lozi dialect, has a long tradition in folk medicine and is known for its medicinal properties, although it is also used as an oil plant. Moreover, its fruit is traditionally used for brewing beer and other alcoholic beverages [15]. *O. serrulata*, locally called munyelenyele, is known worldwide as an ornamental plant valued for its yellow/red flowers with nice scent and fruits resembling Mickey Mouse’s face [16]. In Africa the roots are used as a medicinal plant to treat various illnesses, e.g., bone diseases. The plant also contains biflavonoids with interesting biological properties [17]. However, information about the use of the oil from the seed has not been reported yet.

The aim of this study was to characterize the chemical and qualitative properties (fatty acid profile and content of tocopherols, macro- and microelements) of selected oils and oil cakes obtained from plants naturally occurring in the surroundings of Mongu City, Western Province, Zambia. These results could aid in the commercialization of locally underused species and thus help in the battle with food insecurity and unsafe food practices, while improving the economic and nutritional status of local communities. The species tested in this study could also be introduced to the international market as a novel food. As far as we know, this study is the first to provide a chemical analysis of *Ochna serrulata* oil and discuss its potential commercial uses.

## 2. Results

The seeds of *Parinari curatellifolia* yielded the highest amounts of extractable oil, followed by *Schinziophyton rautanenii* and *Ochna serrulata*. *P. curatellifolia* and *S. rautanenii* were found to be of a drying nature being predominantly composed of unsaturated fatty acids, whereas *O. serrulata* oil had a non-drying character, since it contained saturated fatty acids. *S. rautanenii* oil was principally composed of γ-tocopherol and contained large quantities of vitamin E. The oil of *O. serrulata* was also relatively rich in vitamin E, having almost equal amounts of both α- and γ-tocopherol. Significant differences in chemical composition were observed between oils prepared by the traditional method and the solvent extraction procedure. With only a few exceptions, the cakes of the tested plants had similar proportions of macro- and microelements. In addition, *O. serrulata* oil was richer in chlorophylls and carotenoids, in comparison to the other oils. The yield and contents of fatty acids, tocopherols, tocotrienols, pigments, and macro- and microelements of individual oils are shown in Table 1, Table 2, Table 3, Table 4 and Table 5 respectively. The volatile compounds in *O. serrulata* oil are given in Table 6.

### 2.1. Fatty Acid Content

The seeds of *P. curatellifolia* yielded 71.0% extractable oil containing 55.96% α-eleostearic acid, 20.25% oleic acid, 9.78% linoleic acid, 6.33% palmitic acid, and 5.28% stearic acid. The major fatty acids of *O. serrulata* seed oil (Soxhlet-extracted yield, 35.41%) were oleic acid (37.31%), palmitic acid (37.31%), and linoleic acid (18.66%), with smaller amounts of stearic acid (3.66%). The composition of oil obtained from the market (traditionally prepared) was statistically different (Table 2). The content of oleic acid (46.80%) was higher, while palmitic acid (35.62%) and linoleic acid (10.61%) were lower. Stearic acid was detected at levels of 4.23%. Other fatty acids in both oils were below or slightly above 1%. The oil content of *S. rautanenii* was 56.86% (Soxhlet extraction), of which 46.17% was α-eleostearic acid (tentatively identified according to spectra provided in Figure 1), 24.07% oleic acid, and 10.28% linoleic acid. Relatively large amounts of stearic (9.45%) and palmitic acids (6.29%) were also present. In contrast to the solvent-extracted oil, the traditionally prepared oil was predominantly composed of linoleic acid (40.18%). α-Eleostearic acid represented only 28.58%, and oleic acid only 15.26%, while the contents of stearic and palmitic acids were similar.

### 2.2. Tocopherol and Tocotrienol Content

The tocopherol and tocotrienol content of *P. curatellifolia* oil was relatively low, as it was composed primarily of α-tocopherol (36.37 μg/g) and similar proportions of γ- and δ-tocopherol (6.61 and 6.31 μg/g, respectively). β-Tocopherol and γ-tocotrienol were not detected. The traditionally prepared *O. serrulata* oil had a similar composition. The predominant tocopherol was α-tocopherol at 16.22 μg/g. In comparison to *P. curatellifolia*, this oil was found to have higher quantities of γ-tocotrienol (15.60 μg/g) and β-tocopherol (4.96 μg/g). Interestingly, the solvent-extracted *O. serrulata* oil had a much higher tocopherol content than the one prepared by the traditional method (287.37 μg/g of α-tocopherol and 361.11 μg/g of γ-tocopherol). The highest concentration of tocopherols was found in traditionally prepared *S. rautanenii* oil, which was mainly composed of γ-tocopherol (3236.18 μg/g), together with smaller amounts of δ- (77.69 μg/g) and α-tocopherol (17.32 μg/g). The γ-tocopherol content in Soxhlet-prepared *S. rautanenii* oil was considerably lower (162.95 μg/g), while α-tocopherol content was higher (51.94 μg/g). β-tocopherol and γ-tocotrienol were not detected in either oil. In addition, no δ-tocopherol was found in solvent-extracted *S. rautanenii* oil.

### 2.3. Chlorophylls and Carotenoid Contents

Of all the samples analyzed, *O. serrulata* oil contained the highest amounts of pigments, and the results were comparable to commercial olive oil, which contained 7.02 μg/g of chlorophyll a, 13.82 μg/g of chlorophyll b, and 0.84 μg/g of carotenoids. *O. serrulata* oil obtained by Soxhlet extraction had 16.01 μg/g of chlorophyll a, 7.91 μg/g of chlorophyll b and 7.87 μg/g of carotenoids, whereas oil purchased on the traditional market contained 4.47, 3,95, and 1.32 μg/g of chlorophylls a and b and carotenoids, respectively. Another oil that contained pigments was oil of *P. curatellifolia*, which had 1.24 μg/g of chlorophyll a, 1.73 μg/g of chlorophyll b, and 0.69 μg/g of carotenoids. Chlorophylls and carotenoids were not detected in *S. rautanenii* oils.

### 2.4. Volatiles of the O. serrulata Oil

The oil of *O. serrulata* emits a very specific odor, which was described as foul by many and reminiscent of a decaying human body. The volatile, *p*-cymene, was detected as a major factor in its aroma (relative abundance 52.26%; Figure 2). It appears that the typical odor of *O. serrulata* oil is to a significant degree also determined by m-xylene (9.63%), γ-terpinene (9.07%), o-xylene (7.97%), limonene (7.23%), ethylbenzene (3.23%), α-phellandrene (3.18%), and myrcene (2.47%). Other compounds, e.g., camphene, cyclohexane, propanediol, and propylisovalerate, were detected at levels below 1%. Despite their relatively low abundance, they may still significantly contribute to the odor.

### 2.5. Macro- and Microelements of Cakes

All the analyzed cakes had similar proportions of the elements with some small exceptions (Table 5). Both *P. curatellifolia* and *S. rautanenii* were rich in nitrogen (approx. 90 g/kg), while *O. serrulata* cakes had a significantly lower N content (30 g/kg), but twice as much potassium in comparison to other two species. *S. rautanenii* cake also contained almost twice the concentrations of magnesium (4.5 g/kg) and phosphorus (7 g/kg) compared to *P. curatellifolia* and *O. serrulata*. Only small difference in the sulphur content (2–3 g/kg) were observed in all cakes. Interestingly, *O. serrulata* contained significantly higher levels of aluminium (134.01 mg/kg) and nickel (5.89 mg/kg).

## 3. Discussion

In this study, we determined the chemical composition of oils and performed elemental analysis on oil cakes obtained from trees naturally growing in Zambia’s Western Province. The oil yield from *S. rautanenii* seeds was 3 to 12% higher compared to the seed originating from Namibia or Botswana [12]. The chemical composition of oil is partially known, and studies have found that the oil is predominantly composed of α-eleostearic acid, linoleic acid, oleic acid, palmitic acid, and γ-tocopherol [12,19], which is generally in agreement with our results. The content of α-eleostearic and linoleic acid was more dominant compared to the previous studies and was also depended on the extraction method (Table 2). Furthermore, γ-tocopherol content in traditional prepared oil was twice higher [12]. There are only a few studies documenting the chemical composition of *P. curatellifolia* oil. It was reported that the oil is relatively rich in 18:3 fatty acid isomers and α- and γ-tocopherols (again, in accordance with our results) [20]. Interestingly, the high content of erucic acid (58%) was reported in the oil from seed of *P. curatellifolia* collected in Burundi [21]. This toxic fatty acid has been identified in *P. curatellifolia* in previous studies. This fact can make the use of this oil as food potentially hazardous; further studies are definitely needed to screen the diversity of fatty acid composition of this species within Africa. To the best of our knowledge, this is the first study focusing on the fatty acid, vitamin E, and pigment content of *O. serrulata* oil. With the exception of *S. rautanenii* [12], this study is also the first to present results on elemental analysis of the cakes.

Some differences in chemical composition were observed between traditionally prepared and Soxhlet-extracted oils. The finding that one sample of *S. rautanenii* contained chiefly α-eleostearic acid and the other one linoleic acid has also been reported [22], indicating that various factors might affect the final composition of the oil. It is possible that in a specific processing method, enzymes such as conjugase that are responsible for the conversion of linoleic acid to α-eleostearic acid might be inhibited by excessive heat, resulting in higher concentrations of linoleic acid [18]. Other explanations are related to the observation that trees growing in certain areas, such as Botswana and Zimbabwe, had low levels of α-eleostearic acid [22]. These trees could be subspecies or variants of *S. rautanenii* lacking the enzymatic apparatus required for biosynthesis of α-eleostearic acid. Specific environmental conditions such as water stress and lack of certain macro- and microelements in the soil might also influence the production of this compound. Other inconsistencies were also observed in the case of vitamin E and pigment (chlorophylls *a* and *b*, and carotenoids) content. Again, the processing method together with storage conditions might affect the final composition, since these compounds are prone to degradation when exposed to light and heat [23,24,25]. The vitamin E and pigment content might also be influenced by environmental factors such as drought, excessive sunlight, and genetic variability, where mutants lacked enzymes necessary for biosynthesis [26,27].

The oil, and even the flower and fruit, of O*. serrulata* has a very distinctive odor, which is described by many Zambian locals as foul or reminiscent of a decayed human body. Despite this, the oil is preferred by the locals over *P. curatellifolia* and *S. rautanenii* oil, seemingly due to its superior taste (personal communication, 12 June 2019). It was previously reported by some studies that compounds such as *o*-xylene, *m*-xylene, cyclohexane, and limonene contribute to the typical odor of a decaying human body [28,29]; and as revealed in our study, some of these substances were released from *O. serrulata oil* and were present at relatively high levels, which suggests that these compounds may contribute to the foul odor of this oil.

All the tested species had compositions of macro- and microelements in oil cakes similar to those in soybean meal [30], suggesting that they can be used as animal feed. The cakes of *O. serrulata* had a higher content of nickel. Though rare in animals, symptoms of nickel deficiency include slowed growth, reproductive changes, and altered lipid and glucose levels in the blood [31]. *O. serrulata* oil cake might thus be considered for animals with nickel insufficiency. However, the cake also contains high amounts of aluminum, which might cause toxicity to animals [32].

*P. curatellifolia* is primarily used as a medicinal plant, while *S. rautanenii* serves as a raw material for the manufacture of cosmetic products. However, what is not part of current thinking is that these oils could also be introduced into the food industry. Both *P. curatellifolia* and *S. rautanenii* oil contained significant amounts of α-eleostearic and linoleic acid. Additionally, as previously found, α-eleostearic acid might have some beneficial biological activities such as suppression of tumour growth via induction of apoptosis [33]. The popularity of *O. serrulata* oil among the locals in Zambia’s Western Province because of its resistance to rancidity and its high vitamin E content might be signals that this oil has good potential for use in the food industry. With regard to its fatty acid composition, *O. serrulata* oil resembles palm oil and could be employed in similar ways. One limiting factor for its use in the food industry may be its foul smell, which could be removed, however, by appropriate techniques to render it acceptable to consumers [34].

## 4. Materials and Methods

### 4.1. Plant Material

The plant species used in this study were selected according to documented traditional use as oil crops for cooking purposes (acquired through personal interviews with Lozi people in indigenous communities). Collection of plant material (0.5 kg of fresh fruits from two trees of each species) was performed on farms and in wild areas surrounding Mongu City, Western Province, Zambia, in November 2018. Samples of *Ochna serrulata* (munyelenyele) and *Schinziophyton rautanenii* (mungongo) oils were also purchased from a local market in Mongu (two samples of each oil). Voucher specimens of collected plant material as well as samples of purchased oils were authenticated by Mbao Ngula (projectEDUCATE) and Mukelabai Ndiyoi (University of Barotseland). Herbarium specimens were deposited at the Herbarium of the Czech University of Life Sciences, Prague. Botanical information about the plant species and yields of oils are provided in Table 1.

### 4.2. Chemicals, Standards and Reagents

Supelco 37 component FAME mix and the standards of the volatile compounds propanediol, α-phellandrene, camphene, myrcene, *p*-cymene, and limonene were purchased from Sigma-Aldrich (Prague, Czech Republic). Tocotrienol and the mixed solution of tocopherol standards were from ChromaDex (Los Angeles, CA, USA). Kjeldahl tablets containing 3.5 g K_2_SO_4_ and 3.5 mg Se, sodium hydroxide (NaOH), *n*-hexane of PESTINORM^®^ grade, and methanol HiPerSolv CHROMANORM grade (MeOH) were purchased from VWR International (Stříbrná Skalice, Czech Republic). Tashiro’s indicator was from P-LAB (Prague, Czech Republic). Nitric acid (HNO_3_), sulphuric acid (H_2_SO_4_), benzene, isopropanol, and petroleum ether of analytical grade were from Lach-Ner (Neratovice, Czech Republic). Sea sand was purchased from PENTA (Prague, Czech Republic). Ultrapure HPLC water was acquired from Simplicity UV system (Merck Millipore, KGaA, Darmstadt, Germany).

### 4.3. Sample Preparation

#### 4.3.1. Traditional Oil Preparation Method

The oils were bought at the local market and were prepared by the traditional method of extraction in boiling water. The oily layer was subsequently collected and boiled again to remove the remaining water.

#### 4.3.2. Oil Extraction from Seeds

Oil was extracted from plants by the accelerated Soxhlet method (Randall extraction) in the Ser 148 solvent extractor (Velp Scientifica, Usmate, Italy). Approximately 5 g of homogenized (finely ground) plant material was mixed with the same amount of sea sand to prevent clump formation during extraction. Samples were then transferred to cellulose extraction thimbles and placed in the extractor with petroleum ether as the extraction solvent. The immersion and rinsing phases were continued for 60 min and solvent recovery (evaporation) took 20 min. Two categories of material were obtained by extraction: oils and defatted seed cakes.

### 4.4. Fatty Acid Profile

The identification of fatty acids in each sample of oil was performed using a modification of the alkaline transmethylation procedure developed by Hlisnikovský et al. [35] (ISO 5508:1990): 50 µL of oil was dissolved in 500 µL of benzene and 500 µL of petroleum ether in a 10-mL volumetric flask. One mL of 0.4 M sodium hydroxide solution in MeOH was added and allowed to stand for 20 min. The volumetric flask was filled up to 10 mL with deionized water and the solution was vigorously shaken and allowed to stand at room temperature for 24 h. A 50 µL aliquot of the resulting benzene:petroleum ether layer was diluted in 950 µL of *n*-hexane and used for GC-FID (quantitative) and GC-MS (qualitative) analyses. The GC-FID instrument included a 7890A oven equipped with flame ionization detector (FID) (Agilent, Santa Clara, CA, USA). The separation of fatty acids was performed on an Rt-2560 fused silica column (100 m, 0.25 mm i.d., 0.2 µm; Restek, Lisses, France). The temperature program began with a 2 min hold at 70 °C followed by an increase to 225 °C at a rate of 5 °C/min. The temperature was held at 225 °C for 9 min, then raised to 240 °C at a rate of 10 °C/min and held for 6.5 min (total run, 45.5 min). The flows of the FID detector gases were set at the following values: H_2_, 30 mL/min; air, 400 mL/min; and make-up flow, 30 mL/min. The heater temperature was set at 260 °C. The injection volume was 1 µL with a flow rate of 1.2 mL/min (helium was used as carrier gas), an inlet temperature of 225 °C, and split 1:50. The GC-MS apparatus consisted of a 7890A oven and 5975C mass spectrometer (Agilent Technologies, Santa Clara, CA, USA). The analyses on both instruments were run using the same column, temperature program, injection volume, flow rate, carrier gas, inlet temperature, and split. The temperatures of the ion source and quadrupole were 230 and 150 °C, respectively. Mass spectrum data were acquired in scan mode (range: 40–400 *m*/*z*) and the spectra were identified using the NIST (National Institute of Standards and Technology) mass spectrum library ver. 2.0f. The retention indices of each analyte were calculated from the retention times of *n*-alkanes by linear interpolation as previously described by Kováts [36]. The analyzed fatty acids were identified by comparing their spectra with the spectra of available standards and/or by comparing their retention indices with the NIST database.

### 4.5. Tocopherol and Tocotrienol Contents

Approximately 100 mg of homogenized oil sample was weighed into a 10 mL volumetric flask and filled to volume with isopropanol. The samples were then shaken vigorously and placed in an ultrasonic bath for 10 min (PS 04, Notus Powersonic, Vráble, Slovakia). An aliquot was filtered through a nylon disc filter (0.22 μm) into an amber vial and analyzed by a previously described method for chromatographic separation of individual tocols developed by Lachman et al. [37]. Analysis was carried out using an Ultimate 3000 HPLC system (Thermo Fisher Scientific, Waltham, MA, USA) coupled to an Ultimate 3000RS fluorescence detector. The analytes were separated by isocratic elution on a Develosil 5 μm RP aqueous analytical column (250 mm × 4.6 mm; Phenomenex, Torrance, CA, USA) equipped with a Zorbax SB-C18 (12.5 mm × 4.6 mm, 5 μm; Agilent Technologies, Santa Clara, CA, USA) guard column. The binary mobile phase consisted of methanol and deionized water (97:3, *v*/*v*). The operating conditions were as follows: flow rate, 1 mL/min; time of analysis, 33 min; column temperature, 30 °C; autosampler temperature, 20 °C; injection volume, 10 μL; FLD detection at 292 nm (excitation) and 330 nm (emission). Individual tocols were identified by comparison of retention times and emission spectra with analytical standards. The quantification was based on peak area and external calibration with a concentration range of 0.025–50 μg/mL per analyte and 10-point calibration.

### 4.6. Measurement of Chlorophyll a and b and Carotenoid Content

Chlorophyll *a* and *b* and total carotenoid were determined by a modification of a previously described method [38]. An aliquot of 0.3 g of each oil sample was diluted in 1 mL of cyclohexane, thoroughly vortexed, and transferred to a 96-well microtiter plate (final volume 200 μL). The absorbance at 470, 645, and 662 nm was measured on a Synergy H1 multi-mode microplate reader (BioTek, Winooski, VT, USA). The −acquired absorbance data were converted to concentration (μg/g) according to the following formulas: chlorophyll *a* (Equation (1), chlorophyll *b* (Equation (2)) and total carotenoids (Equation (3)):(1)$$C_{a}=(11.24\;\times {\;A}_{662})\;-\;(2.04\;\times {\;A}_{645})$$
(2)$$C_{b}=(20.13\;\times {\;A}_{645})\;-\;(4.19\;\times {\;A}_{662})$$
(3)$$\mathrm{total}\;\mathrm{carotenoids}\;=\frac{\left[\left(1000\times {\;A}_{470}\right)-(1.63\times {\;C}_{a})-\left(63.14\times {\;C}_{b}\right)\right]}{1000}\;$$

The dilution factor was included in the final calculation of the given concentrations. Extra virgin olive oil (Franz Joseph Kaiser, Zlín, Czech Republic) was used for reference.

### 4.7. GC-MS Analysis of Volatiles in O. serrulata Oil after Solid-Phase Microextraction (SPME)

Since *O. serrulata* oil possessed a distinctive odor described by many as foul, it was subjected to SPME/GC-MS analysis to determine which compounds might be responsible for the unusual odor. The instrument consisted of an Agilent GC7200 oven equipped with 7890B qTOF (Santa Clara, CA, USA) and CombiPAL autosampler (CTC Analytics AG; Zwingen, Switzerland). The volatiles were adsorbed onto an SPME fibre filter coated with a combined DVB/CAR/PDMS phase (Supelco, Bellefonte, PA, USA). The oil samples were conditioned for 5 min at 40 °C and sorption took place at the same temperature for 2 min. The volatiles were desorbed in the GC inlet for 5 min at 250 °C in splitless mode (times and temperatures were based on manufacturer’s instructions). Separation of individual constituents was performed on an HP-5MS column (30 m, 0.25 mm i.d., 0.25 μm; Agilent). Helium was used as carrier gas at a flow rate of 1 mL/min. The temperature program of the oven began with a 5-min hold at 40 °C. The temperature was then raised to 180 °C at a rate of 5 °C/min, held there for 1 min, and then raised to 280 °C at 30 °C/min for a total run time of 37 min. The temperature of the quadrupole was maintained at 230 °C and the ion source at 230 °C. The compounds were measured in scan mode over a range of 55–700 Da. The analyzed volatiles were identified by comparing their spectra with the spectra of standards and/or by comparing their retention indices (calculated according to Kováts) with the NIST database.

### 4.8. Determination of Nitrogen Content in the Cakes

Nitrogen content in the cakes was determined with the modified titration method developed by Kjeldahl [39]. One gram of sample was mixed with 20 mL of concentrated H_2_SO_4_ containing two Kjeldahl catalyst tablets and digested at 420 °C until the solution was clear. The samples were cooled to room temperature, mixed with 60 mL of distilled water and 70 mL of 40% NaOH, and subjected to hydrodistillation. The resulting ammonia was collected in 30 mL of a solution of 1% boric acid with a few drops of Tashiro’s indicator. The concentration of ammonia was determined by titration with 0.2 N H_2_SO_4_.

### 4.9. Identification of Other Macro- and Microelements

The determination of minerals was carried out after acid digestion of samples performed according to a modification of the published method [40]. Due to the high fat content of the samples, the reaction time was substantially increased to ensure total digestion. Initially, 0.5 g of sample in 10 mL of 65% HNO_3_ was kept overnight at laboratory temperature in closed, but not sealed, Teflon vessels (Savillex, Eden Prairie, MN, USA). Afterwards, the Teflon vessels were sealed, and the mixture was heated at 120 °C for 2 h. The digested solutions were then quantitatively transferred to 50 mL volumetric flasks and filled up to the mark with deionized water. Before analysis, the solutions were filtered through 0.45 µm nylon disks. An inductively coupled plasma optical emission spectrometer (DUO iCap7000, Thermo Scientific, Waltham, MA, USA) was operated at 1.15 kW with nebulizer and auxiliary gas flow rates set at 0.5 and 1 L/min, respectively. The quality of digestion and analysis were controlled using blanks and the standard reference materials, NIST SRM 1575a (pine needles) and NCS DC 73,351 (tea).

### 4.10. Statistical Analysis

All analyses were performed on means from three independent experiments, each in triplicate. Statistical evaluation of data was performed in Statistica version 12 (StatSoft, CZ). ANOVA followed by Tukey’s test, or an independent *t*-test was performed, as appropriate. The results are expressed as means plus-or-minus standard deviation (mean ± SD).

## 5. Conclusions

Currently, the negative effects of unified and intensive agriculture (e.g., loss of biodiversity, adaptability of crop to climate change, growing reliance of some countries on commodity import) are of growing concern. One of the solutions to mitigate them is to support biodiversity in agriculture by revealing and promotion of the potential of traditional but nowadays underused crop. The study provides much-needed new information on chemical composition of oils and cakes obtained from oil-bearing plants from Western Province, Zambia. The character of fatty acid composition of *S. rautanenii* and *P. curatellifolia* oils defined them as drying oils, whereas *O. serrulata* appeared to be of a non-drying (rancid-resistant) character. Based on the obtained nutritional data, it is evident that all the tested species showed the potential of being introduced into a wide spectrum of industries, especially food industry as salad or cooking oil or food supplement or in technical industry for the manufacturing of soaps, paints, and varnishes, and in the pharmaceutical industry as carrier oils. Because *O. serrulata* contains relatively large quantities of α-tocopherols and pigments, and is resistant to rancidity, it especially deserves greater research attention. The nutrient-dense oil cakes of the tested species are also promising sources of animal feed. Further studies are needed to reveal other qualitative properties, such as iodine, peroxide, saponification values, etc., of these oils and verify their safety for human and animal consumption. Last but not least, more data to evaluate environmental and socio-economic aspects of their cultivation, harvest and processing are needed.

## Figures and Tables

**Figure 1 molecules-26-05210-f001:**
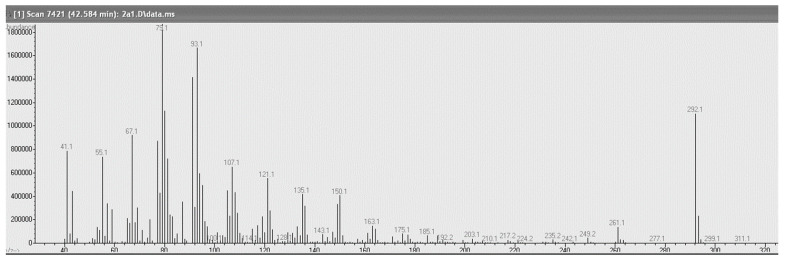
The GC-MS spectra of α-eleostearic acid methyl ester. The masses 41, 55, 67, 79, 93, 107, 121, 135, 150, 163, 185, 261, and 292 and their relative intensities are very specific for α-eleostearic acid methyl ester as observed by Dyer, 2003 [18].

**Figure 2 molecules-26-05210-f002:**
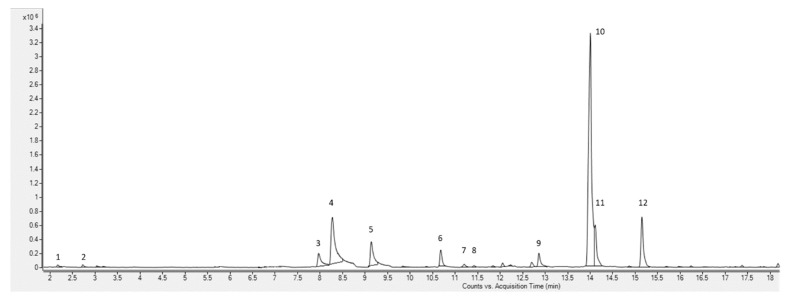
The SPME/GC-qTOF chromatogram of *O. serrulata* oil: 1, propanediol; 2, cyclohexane; 3, ethyl benzene; 4, *m*-xylene; 5, *o*-xylene; 6, α-phellandrene; 7, camphene; 8, propylisovalerate; 9, myrcene; 10, *p*-cymene; 11, limonene; 12, γ-terpinene.

**Table 1 molecules-26-05210-t001:** General information and oil yields of the tested oil plants.

Species	Voucher Specimen No.	Family	Vernacular Name (*Lozi*)	Oil-Bearing Part	Oil Use	Oil Yield (%)
*Ochna serrulata* Walp.	[JT002]	Ochnaceae	*Munyelenyele*	Seeds	Cooking, soap making	35.41 ± 4.43
*Parinari curatellifolia* Planch. ex Benth.	[JT001]	Chryso- balanaceae	*Mubula*	Seeds	Cooking, paint and varnish preparation	71.00 ± 0.00
*Schinziophyton rautanenii* (Schinz) Radcl.-Sm.	[JT003]	Euphorbiaceae	*Mungongo* ^1^	Seeds	Cooking, cosmetic products	56.86 ± 0.59

^1^ Also known as Manketti, but this name is probably derived from a different language from Lozi.

**Table 2 molecules-26-05210-t002:** Fatty acid composition of the oils.

Fatty Acid		Oil				
(%)		*O. serrulata*		*P. curatellifolia* ^1^	*S. rautanenii*	
		Traditional Preparation ^2^	Soxhlet Extraction ^3^	Soxhlet Extraction ^3^	Traditional Preparation ^2^	Soxhlet Extraction ^3^
myristic	C14:0	0.11 ± 0.0001 ^a^	0.10 ± 0.002	0.02 ± 0.001	0.03 ± 0.001	ND
pentadecaonic	C15:0	0.05 ± 0.0002	ND	ND	ND	ND
palmitic	C16:0	35.62 ± 0.05 ^a^	37.31 ± 0.12	6.33 ± 0.02	8.83 ± 0.04 ^a^	6.29 ± 0.06
palmitoleic	C16:1 (9c)	0.14 ± 0.002	0.36 ± 0.004	ND	0.04 ± 0.0003	ND
heptadecanoic	C17:0	0.32 ± 0.001	0.30 ± 0.02	ND	ND	ND
heptadecenoic	C17:1 (10c)	0.09 ± 0.001	0.09 ± 0.01	ND	ND	ND
stearic	C18:0	4.23 ± 0.07 ^a^	3.66 ± 0.08	5.28 ± 0.42	6.13 ± 0.01 ^a^	9.45 ± 1.96
elaidic	C18:1 (9t)	ND	ND	0.03 ± 0.01	ND	ND
oleic	C18:1 (9c)	46.80 ± 0.22 ^a^	37.31 ± 0.22	20.25 ± 0.09	15.26 ± 0.02 ^a^	24.07 ± 0.65
vaccenic	C18:1 (11c)	0.79 ± 0.23 ^a^	1.21 ± 0.04	0.49 ± 0.01	0.46 ± 0.004	0.42 ± 0.01
linolelaidic	C18:2 (9t,12t)	ND	ND	ND	ND	0.89 ± 0.05
linoleic	C18:2 (9c,12c)	10.61 ± 0.02 ^a^	18.66 ± 0.28	9.78 ± 0.09	40.18 ± 0.1 ^a^	10.28 ± 0.28
α-eleostearic ^4^	C18:3 (9c,11t,13t)	ND	ND	55.96 ± 0.21	28.58 ± 0.15 ^a^	46.17 ± 1.48
γ-linolenic	C18:3 (6c,9c,12c)	0.01 ± 0.005	ND	ND	ND	ND
α-linolenic	C18:3 (9c,12c,15c)	0.34 ± 0.01 ^a^	0.40 ± 0.002	ND	0.04 ± 0.0003	ND
arachidic	C20:0	0.15 ± 0.002	0.16 ± 0.002	0.31 ± 0.01	0.16 ± 0.001 ^a^	0.41 ± 0.01
eicosaenoic	C20:1 (11c)	0.27 ± 0.003 ^a^	0.23 ± 0.004	0.91 ± 0.04	0.30 ± 0.003 ^a^	0.78 ± 0.02
eicosadienoic	C20:2 (11c,14c)	ND	0.01 ± 0.002	ND	ND	ND
eicosapentaenoic (EPA)	C20:5 (5c,8c,11c,14c,17c)	0.04 ± 0.01 ^a^	0.07 ± 0.03	ND	ND	ND
behenic	C22:0	ND	0.03 ± 0.01	0.04 ± 0.002	ND	ND
tricosanoic	C23:0	ND	ND	0.11 ± 0.01	ND	ND
lignoceric	C24:0	ND	0.13 ± 0.1	ND	ND	ND
Sum of SFA (%)		40.48	41.67	11.98	15.15	16.15
Sum of MUFA (%)		48.09	39.19	21.68	16.05	25.27
Sum of PUFA (%)		11.00	19.14	65.74	68.80	57.34
MUFA:SFA ratio		1.2	0.9	1.8	1.1	1.6
PUFA:SFA ratio		0.3	0.5	5.5	4.5	3.6

^1^ *P. curatellifolia* (mubula) oil was not available in the market at the time of plant material collection. ^2^ The oil was bought at the local market in Mongu, Western Province, Zambia and was prepared by the traditional method, i.e., melting out with boiling water. ^3^ The oil was extracted from seeds by Soxhlet extraction (Materials and Methods). ^4^ α-eleostearic acid was not part of the FAME mix; thus, it was identified tentatively according to MS spectra (Figure 1). ^a^ Letters indicate statistically significant differences (*p* < 0.01) in FA content in the same plant oil obtained by different extraction methods. Note: results are expressed as relative percentage obtained by FID peak area normalization.

**Table 3 molecules-26-05210-t003:** Tocopherol and tocotrienol contents of the oils.

Compound	LOD	LOQ	Oil				
(μg/g)	(μg/g)	(μg/g)	*O. serrulata*		*P. curatellifolia* ^1^	*S. rautanenii*	
			Traditional Preparation ^2^	Soxhlet Extraction ^3^	Soxhlet Extraction ^3^	Traditional Preparation ^2^	Soxhlet Extraction ^3^
α-tocopherol	5.00	15.2	16.22 ± 4.81 ^a^	287.37 ± 16.63	36.37 ± 8.36	17.32 ± 0.84 ^a^	51.94 ± 12.90
β-tocopherol	1.50	4.5	4.96 ± 1.56	˂LOQ	˂LOD	˂LOD	˂LOD
γ-tocopherol	1.50	4.5	7.94 ± 1.20 ^a^	361.11 ± 19.30	6.61 ± 0.93	3236.18 ± 43.99 ^a^	162.95 ± 11.52
δ-tocopherol	1.50	4.5	6.52 ± 1.83	˂LOQ	6.31 ± 0.71	77.69 ± 0.88	˂LOD
γ-tocotrienol	1.20	3.6	15.60 ± 5.36	˂LOQ	˂LOD	˂LOD	˂LOD

^1^ *P. curatellifolia* (mubula) oil was not available in the market at the time of plant material collection. ^2^ The oil was bought at the local market in Mongu, Western Province, Zambia and was prepared by the traditional method, i.e., melting out with boiling water. ^3^ The oil was extracted from seeds by Soxhlet extraction (Materials and Methods). ^a^ Letters indicate statistically significant difference (*p* < 0.01) in tocopherol content in the same plant oil obtained by different extraction method. LOD = limit of detection, LOQ = limit of quantification.

**Table 4 molecules-26-05210-t004:** Pigment content in the analyzed oils.

Compound	Oil					
(μg/g)	Olive	*O. serrulata*		*P. curatellifolia* ^1^	*S. rautanenii*	
		Traditional Preparation ^2^	Soxhlet Extraction ^3^	Soxhlet Extraction ^3^	Traditional Preparation ^2^	Soxhlet Extraction ^3^
Chlorophyll *a*	7.02 ± 1.39	16.01 ± 1.42 ^a^	4.47 ± 0.83	1.24 ± 0.09	ND	ND
Chlorophyll *b*	13.82 ± 2.86	7.91 ± 2.69 ^a^	3.95 ± 1.61	1.73 ± 0.13	ND	ND
Carotenoids	0.84 ± 0.55	7.87 ± 0.04 ^a^	1.32 ± 0.03	0.69 ± 0.11	ND	ND
Sum of pigments	21.69 ± 4.77	31.79 ± 4.07	9.74 ± 2.43	3.66 ± 0.28	-	-

^1^ *P. curatellifolia* (mubula) oil was not available in the market at the time of plant material collection. ^2^ The oil was bought at the local market in Mongu, Western Province, Zambia and was prepared by the traditional method, i.e., melting out with boiling water. ^3^ The oil was extracted from seeds by Soxhlet extraction (Section 4). ^a^ Letters indicate statistically significant difference (*p* < 0.01) in pigment content between the samples.

**Table 5 molecules-26-05210-t005:** Macro- and microelements in the cakes obtained after Soxhlet oil extraction.

Element	LOD		Plants	
(mg/kg DW)	(mg/kg)	*O. serrulata*	*P. curatellifolia*	*S. rautanenii*
Al	96.51	134.01 ± 20.69	ND	ND
Ba	16.84	31.54 ± 1.94	32.16 ± 0.17	36.25 ± 1.99
Ca	19.71	2850 ± 6.5	2753 ± 54	3364 ± 8.6 ^a^
Cr	1.86	ND	ND	2.53 ± 0.62
Cu	14.93	23.03 ± 2.68	28.78 ± 0.82	32.46 ± 0.73
K	233.53	9622 ± 119 ^a^	4714.25 ± 246	6243 ± 134
Mg	7.16	1560 ± 2.6 ^a^	2808 ± 72.2 ^b^	4490 ± 97.5 ^c^
Mn	0.37	25.56 ± 0.26	23.33 ± 1.85	37.76 ± 0.18 ^a^
N	-	30181 ± 119 ^a^	93312 ± 133 ^b^	91298 ± 562 ^c^
Na	64.59	232.75 ± 33.43	249.09 ± 38.2	284.93 ± 12.22
Ni	1.00	5.89 ± 0.18 ^a^	1.95 ± 0.36	2.45 ± 0.43
P	16.87	2724 ± 36 ^a^	4867. ± 138.4 ^b^	7140 ± 87.2 ^c^
S	30.18	1890 ± 49.5 ^a^	3085 ± 139.5	2755 ± 17.5
Zn	15.01	43.02 ± 1.01	42.56 ± 2.16	40.98 ± 2.26

^a–c^ Letters indicate statistically significant difference (*p* < 0.01) in element content between the samples.

**Table 6 molecules-26-05210-t006:** Relative proportions of volatile compounds emitted by *O. serrulata* oil.

Compound	RT ^1^	Mean ± SD	Kováts Indices (KI)		Mode of Identification	Characteristic Odor ^3^
	(min)	(%)	Calculated ^2^	in Literature		
propanediol	1.66	0.32 ± 0.01	691	732	standard, NIST	odorless
cyclohexane	2.73	0.43 ± 0.04	720	667	NIST, KI	sweet, gasoline-like; chloroform-like
ethylbenzene	7.97	3.23 ± 0.37	859	855	NIST, KI	similar to that of gasoline
*m*-xylene	8.27	9.63 ± 0.03	867	866	NIST, KI	sweet
*o-*xylene	9.14	7.97 ± 0.20	890	889	NIST, KI	sweet
α-phellandrene	10.68	3.18 ± 0.09	931	1012	standard, NIST	black pepper
camphene	11.12	0.55 ± 0.03	945	954	standard, NIST	fresh herbal, woody ^2^
propylisovalerate	11.41	0.26 ± 0.10	952	949	NIST, KI	fruity
myrcene	12.86	2.47 ± 0.22	990	992	standard, NIST	earthy, fruity, and clove-like; pungent in higher concentrations.
*p*-cymene	13.99	52.26 ± 0.22	1023	1023	standard, NIST	sweet, soft, fresh, lemon, bergamot
limonene	14.09	7.23 ± 0.16	1027	1024	standard, NIST	lemon-like
γ-terpinene	15.15	9.07 ± 0.08	1058	1062	NIST, KI	terpentiny, sweet, citrus, with tropical and lime nuances

^1^ Retention time. ^2^ KI values were calculated according to retention times of C7-C30 *n*-alkanes analyzed by the GC-MS method described in Section 4.4. ^3^ Information obtained from the following sources: www.thegoodscentscompany.com/, pubchem.ncbi.nlm.nih.gov, https://www.perfumersworld.com. Note: results are expressed as relative percentage obtained by peak area normalization.

## Data Availability

Data is contained within the article.

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
