# Peer review of "The Chemical Composition of Oils and Cakes of *Ochna serrulata* (Ochnaceae) and Other Underutilized Traditional Oil Trees from Western Zambia"

_molecules, 2021, doi:10.3390/molecules26175210_

Round 1
Reviewer 1 Report
In my opinion the present manuscript is now suitable for the publication. Just a small remark, in query 17 the authors response was the follow: "The time was based on manufacturer’s instructions (https://www.sigmaaldrich.com/CZ/en/technical-documents/technical-article/analytical-chemistry/solid-phase-microextraction/nitinol-core-spme-fibers). This fact is now acknowledged in the materials and methods."
In the examples reported in the SIGMA website are reported different samples therefore when a new sample is considered it should be better to perform a new optimization mehtod fot the extraction parameters.
Author Response
Thank you for your comment. The optimization of the SPME method (especially the sorption time) was done. Unfortunately, there was misunderstanding between the colleagues who provided information and Dr. Tauchen
Reviewer 2 Report
The manuscript reports data on chemical composition of oils and cakes of three traditional oil plants in Zambia. Although the measurement methods are routine methods, the research object is unknown, and the research results will help the public and scientific community understand the species. I agree that this study has some novelty.
- The word “neglected” in the Title is not suitable. More than 10 literatures on the oils of Parinari curatellifolia or Schinziophyton rautanenii could be found in SciFinder.
- In the Abstract, no sentence mentions to the other two plant Parinari curatellifolia or Schinziophyton rautanenii. What’s the aim of the study?
- For keywords, pertinent keywords specific to the article but quite common in the subject discipline should be given. The five key words in the paper do not meet the requirements of the journal. I suggest to give the species of the three oil plants at least.
- Standard professional vocabulary is required, please revise the words “macro- and micro-elements, pigmented compounds, cakes” and so on.
- The first sentence in the Introduction is incorrect. Apparently, far more than 200 species of vascular plants are cultivated by farmers.
- In Table 1, which method was used to obtain the Oil yield?
- A Rt-2560 fused silica column (100 m) was used for fatty acids analysis. Normally a 30m GC column is good enough, why the author used a 100m length column?
- In the SPME method, the choice of fiber type has a decisive influence on the GC-MS results. Why the authors used PDMS/DVB fiber?
- The RI values of the VOCs should be calculated by analyzing the C7–C30 n-alkanes under the same GC–MS conditions as samples. No related information of the standers was seen in the methods.
- There are some reports on the oils of Parinari curatellifolia or Schinziophyton rautanenii. Please quote and supplement relevant discussions rather than simply describe the authors’ own data.
- The preface needs to be refined, and to supplement the overview of the three oil plants.
Reviewer 3 Report
Dear authors,
please enclose you will find the comments to the manuscript
Best regards

Author Response
Thank you for your comments and recommendations. The abstract, keywords and introduction were changed. The changes are indicated in yellow colour. Information about the plants were added to the introduction. Please see the lines 22-36 and 63-96.
Reviewer 4 Report
The paper entitled ‘The chemical composition of oils and cakes of Ochna serrulata (Ochnaceae) and other neglected traditional oil trees from western Zambia’ presented by Frankova et al. is based on interesting traditional oil trees which was not studied in detail up to now. Authors focused on chemical composition (fatty acid content, tocopherol and tocotrienol content, chlorophylls and carotenoid contents, macro- and micro-elements) of oils and cakes. In my opinion, the paper is well-prepared. The methodology is adequate and well-organized. Results are presented in form of tables and figure. Statistical analysis was performed. Discussion is presented in detail. In my opinion the paper can be accepted for publication in Molecules but the following suggestion should be considered:
- References are not up-to-date. Please use more current references.
- Conclusions could be more expanded
Author Response
Thank you for your comments and recommendations. Some new references were added. The oldest references given (extraction method, botanical description) still contain valid information. The conclusions were expanded, changes are indicated in yellow.
Round 2
Reviewer 2 Report
The quality of this article has been improved to a certain extent.
This manuscript is a resubmission of an earlier submission. The following is a list of the peer review reports and author responses from that submission.
Round 1
Reviewer 1 Report
The revision looks good and I would recommend to accept for publication.
Reviewer 2 Report
Comments/suggestions: 1) The last sentence of abstract “To the best of our knowledge, this is the first study that deals with chemical analysis of O. serrulata outlining its possible commercial use.” should be removed. Phytochemical and Biological Studies of Ochna Species 2012, Chemistry & Biodiversity by Anil Kumar Reddy Bandi EFFECT OF SOME FERTILIZATION TREATMENTS ON GROWTH AND CHEMICAL COMPOSITION OF OCHNA SERRULATA (HOCHST.) WALP. SHRUBS, Amal S. El-Fouly 2) Authors in Abstract are mentioned that “This study presents results on chemical composition of oils and cakes of traditional oil plants used by descendants of Lozi people in surroundings of Mongu, Western province, Zambia, with special emphasis being placed on the so far undescribed Ochna serrulata (Ochnaceae).” This sentence is not clear. 2) Table 1. The title is missing. What is the purpose of this Table. 3) In the purpose of the study Authors should mention the selected oils and cakes obtained from plants. 4) The lines 86-90 is a promise and nor the purpose of the study. Therefore, authors should state clearly the aims, the originality and the necessity of their research. 5) The text of the Results should be rewritten as it has too much ambiguity and doubtfulness. 6) 2.1. paragraph repeats the results of Table 2. Moreover, Table 2 presents the fatty acid proportions or composition? Statistical analysis of Table 2 results should revise to a better presentation. 7) 2.2. paragraph repeats the results of Table 3. Statistical analysis of Table 3 results should revise to a better presentation. 8) 2.3. paragraph repeats the results of Table 4Statistical analysis of Table 4 results should revise to a better presentation. 9) The same revision should be done for 2.4 and 2.5 paragraphs. 10) Figure 1 should be mention and discuss in the text. 11) It is written: “This study dealt…” Please explain. 12) The authors continuously write that their work is done for the first time, without presenting innovative findings beyond simple measurements. 13) What the purpose of GC-FID and GC-MS analyses, as only GC-FID results are given and discussed. The fatty acid qualitative and quantitative determination should be added in details.Reviewer 3 Report
The oil extraction method should be better explained. Is the solvent (petrol ether) evaporated after extraction, the yield of extraction should be reported.
Tables footnote must report the units of data reported, as for example “results are expressed as relative percentage obtained by FID peak area normalization”
The authors should explain how they choose the extraction time of SPME extraction. Is it based on literature data or they performed optimization experiments?
Lines 177-185: the authors carried out only a qualitative analysis since the SPME extraction show very strong variability. The amount of each compound in the fiber is related to the affinity of such compound with the stationary phase of the fiber, from the time of fiber exposition to the head-space (several competition phenomena occurs during extraction) and finally from the amount of the target compound in the head-space. In addition mass detector show a wide range of response factors which is related to the structure of the target compound. Therefore the this section should be completely revised.
The English should be revised by a native English speaker throughout the text.